# Diet Influence on Sperm Quality, Fertility, and Reproductive Behavior in Karakul of Botoșani Rams

Constantin Pascal [1], Ionică Nechifor [2], Marian Alexandru Florea [2,*], Claudia Pânzaru [1,*], Daniel Simeanu [3] and Daniel Mierliță [4]

1. Department of Animal Resources and Technologies, Faculty of Food and Animal Sciences, Iași University of Life Sciences, 700490 Iasi, Romania; pascalc@uaiasi.ro
2. Research and Development Station for Sheep and Goat Breeding, Popăuți-Botoșani, 710004 Botoșani, Romania; nechifor.ionica@karakul-moldoovis.ro
3. Department of Control, Expertise and Services, Faculty of Food and Animal Sciences, "Ion Ionescu de la Brad" University of Life Sciences, 8 Mihail Sadoveanu Alley, 700489 Iasi, Romania; dsimeanu@uaiasi.ro
4. Faculty of Environmental Protection, University of Oradea, 26 General Magheru Boulevard, 410048 Oradea, Romania; daniel.mierlita@uoradea.ro
* Correspondence: florea.marian@karakul-moldoovis.ro (M.A.F.); panzaruclaudia@uaiasi.ro (C.P.)

**Abstract:** This study aims to analyze the influence of an improved diet with vitamins and minerals (VM) on the live weight, body condition, quality of sperm, behavior, and fertility of rams. The biological material comprised two groups of rams (L1—control and L2—VM supplemented), each consisting of 15 individuals. After a complete one-year cycle, they received different dietary treatments at the beginning of the preparation for the reproduction period. Although in the onset of the mounting period (SM), no significant differences were observed for live weight ($p > 0.05$), providing supplemental feeding of a VM complex allowed a better capitalization of body reserves, and, consequently, the rams' groups differed significantly by the end of mating season (FM), for live weight (+4.1%; $p < 0.001$) and body condition score (+15.9%; $p < 0.05$). Adding vitamins and minerals to the L2 diet also improved sperm color ($p < 0.001$), sperm concentration (+11.8%; $p < 0.01$), live spermatozoa (+2.6%; $p < 0.001$), and decreased abnormal spermatozoa proportion (−7.0%; $p < 0.01$). The increase in the scrotum circumference in L2 (+4.57%) suggests that VM supplements improved testosterone secretion, spermatogenesis, and ejaculate volume (+10.20%; ($p < 0.001$), with a positive impact ($p < 0.001$) on mating behavior, on the gestation installation (+11.2%) and on the number of obtained lambs (+14.0%), as well as on the key economic indicators (+13.8% incomes per ram).

**Keywords:** rams; fertility; Botoșani Karakul breed; semen; spermatozoa

## 1. Introduction

In Central and Eastern Europe, the impact of climate change has intensified significantly lately. In a changing environment, the pasture dries up in the summer, and small ruminants are affected even before the breeding season. Nutrition can have a more significant influence on rams' performance and testicular size than the photoperiod. The purpose of nutrition in breeding them is an important aspect that contributes to achieving positive results. Studies indicate that spermatogenic activity, sperm content in the epididymis, and many other traits are higher in rams fed ad libitum. In a study conducted by Oldham et al., 1978 [1], the data obtained show that the efficiency of a balanced diet on testicular activity is important because rams that received a balanced diet had more intense activity and produced more spermatozoa than those with inadequate nutrition.

Research conducted on sheep populations in different arid regions indicates significant effects caused by nutritional deficiencies or climate changes, with a major impact on reproduction. Some of these results highlight the importance of proper and balanced feeding because it positively influences the onset of puberty, ovulation rate, embryo survival,

anestrus length, and the response of sheep to interaction with rams [1,2]. Adequate nutrition can also influence certain traits, such as the fertilization capacity of rams [3], semen quality, and reproductive performance, leading to better economic outcomes [4]. When provided with inadequate food and exposed to climatic factors outside the comfort range, not only will the reproductive capacity be affected, but also the quantity and quality of sperm [5]. Therefore, through nutrition, the goal is to maintain or improve the maintenance of rams to achieve an optimal level of their body condition and best performance. The results of some studies have indicated the beneficial effects of feeding Targhee rams with Zn supplements, although the source of Zn has elicited varied responses in the measured performance characteristics [6].

The fertility of rams represents a management issue influenced by both their health and the quality of their feeding. Bioactive feed components can provide protection against oxidative and inflammatory damage of the male reproductive system. Additionally, this action will improve spermatogenesis and other specific traits related to the reproductive function of rams [6,7]. In the absence of activities supporting the reproductive function, reproductive pathological conditions may arise as a significant barrier to achieving positive results.

At the farm level, the management of rams is complex and heavily influenced by local factors, breed, farm, season, and the farmer's experience [8,9]. Effective management involves genetics, health, nutrition, behavior, economics, and physiological and anatomical changes occurring throughout or outside the breeding season [10,11].

Often, physical, chemical, biological, or technological aspects are not adequately addressed, thus leading to negative effects. Prolonged exposure of rams and ewes to the direct influence of excessively high temperatures can be a stress factor, affecting their behavior, their reproductive function, and sperm production levels [12]. Heat stress induces various physiological, metabolic, endocrine, and molecular disruptions in the body as a response to the effort in maintaining thermal balance [13,14]. To regulate body temperature and ensure survival, certain endocrine changes occur within the body that have a negative association with ram fertility [12]. Prolonged heat stress leads to reduced testosterone concentrations, negatively influencing the spermatogenesis and sexual behavior of rams [10,12,15].

Even a half-degree change in body temperature can reduce spermatogenesis and/or libido [16]. In such circumstances, it is recommended to implement the following activities that enhance thermal comfort during critical periods to mitigate the effects of heat stress: housing the rams in shaded and well-ventilated enclosures, providing antioxidant-enriched diets, and maintaining an appropriate feeding regimen [17].

The purpose of this research was to study the effects of certain technological factors that can be easily monitored on farms and might influence reproductive activity of the Botoșani Karakul breed. Another goal was to identify effective technological methods to mitigate the effects of thermal discomfort, inadequate maintenance, and proper nutrition to ensure improved values for ram fertility. The main objective of this research was to enhance the reproductive performance of rams by implementing a diet that includes a VM (vitamins and minerals) supplement.

## 2. Materials and Methods

### 2.1. Ethical Approval

The activities scheduled during the research were consistent with the ethics of scientific research. When collecting semen and blood samples, considerations from the Ethics and Experiments on Animals were taken into account, and procedures were applied in accordance with the Ethics Committee's regulations of the research and development unit, as well as the recommendations of the Institute of Diagnosis and Animal Health in Romania. Likewise, in the case of handling and collecting biological samples, all ethical requirements were followed to ensure that animals subjected to experimental procedures were not put in discomfort or made to undergo any painful treatments (941/04.10.2021).

### 2.2. The Research Area and Climatological Conditions

The research area is located in the northeastern part of Romania, being located at 47°44′55″ north latitude and 26°40′10″ east longitude. In this region (Figure 1), climate changes have become obvious, and there is a significant increase in the number of days with daily average temperatures exceeding +35 °C compared to multiannual average (https://www.meteoblue.com, accessed on 12 July 2023).

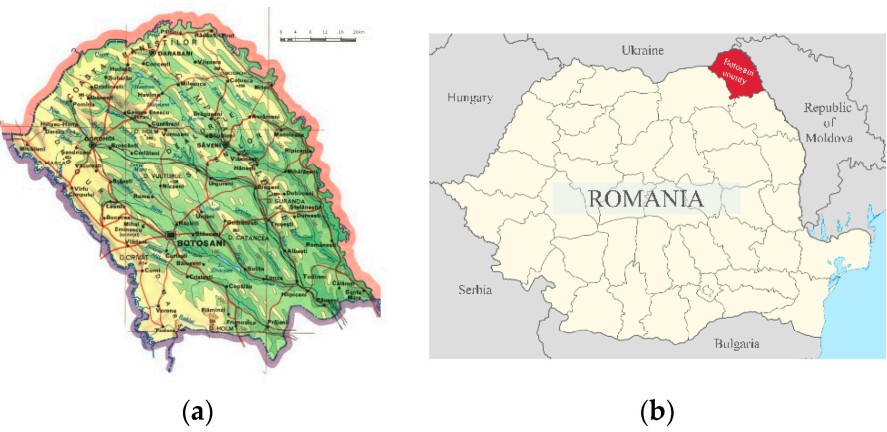

|(**a**)|(**b**)|

**Figure 1.** Representation of the research area (**a**) and its location on the map of Romania (**b**).

Current climate changes lead to an increase in hot and dry days during the summer, while winters become quite mild and devoid of snow. In this context, climate changes have a strong and negative impact not only on the environment but also on sheep, affecting their behavior and causing certain disruptions to their physiology.

### 2.3. Animal, General Procedures, and Data Management

The research was conducted on the sheep population within the Research and Development Station for Sheep and Goats in Popăuți-Botoșani. The biological material used consisted of two groups of adult rams (L1, L2), each comprising 15 rams aged between 2 and 5 years. At the start of the breeding activity, the live weight for L1 was 86.06 ± 2.16 kg, while for L2, it recorded a value of 90.13 ± 2.19 kg. All research activities were carried out over a complete reproduction cycle of one year, and the maintenance and feeding conditions applied to both groups were identical, only differing during the period of sexual rest. The experiment was based on the administration of a VM supplement only to L2 during the preparation period (MP) and the sexual activity period (SM), corresponding to the natural breeding season during autumn (September–November).

During the MP period, the batch L2 received an improved diet. A complex of vitamins and minerals (VM) was additionally provided to support the reproductive activity (Table 1). Daily nutrition included certain minerals (Zn, Fe, Mn, Cu, Co, I, Se) and vitamins (A, D, E). The diets provided to both groups aimed to support body condition, and the addition of VM to L2 ensured the necessary macro- and micronutrients to support the reproductive function and fertility of the rams. In accordance with the experimental protocol, the research also included an evaluation of how body condition influenced the quantity and quality of seminal material in rams, as well as the behavior and fertility of the rams.

Water was provided from safe and abundant sources, and the preparation for mating and the actual mating activity took place during the natural breeding season (September–October). Since the Botoșani Karakul breed is developed and improved for pelts production, the used reproductive system is natural mating (each ram mounts 35 ewes per season). Based on the data obtained in EM, several quality traits of sperm, fertility, and the economic efficiency of the proposed experimental model were determined.

**Table 1.** Structure of the diet throughout the experimental period.

| Dietary Ingredients (kg) | Diet for L1 | | | | Diet for L2 | | | |
|---|---|---|---|---|---|---|---|---|
| | Rest during Summer | Rest during Winter | Preparation for Mating | Mating | Rest during Summer | Rest during Winter | Preparation for Mating | Mating |
| Alfalfa hay | - | 0.410 | 0.210 | - | - | 0.410 | 0.210 | - |
| Barley straw | 1.430 | 1.610 | 1.340 | 1.420 | 1.430 | 1.610 | 1.340 | 1.420 |
| Hillside pasture | 2.400 | - | - | 2.800 | 2.400 | - | - | 2.800 |
| Carrots | - | - | - | 0.200 | - | - | - | 0.200 |
| Barley | 0.095 | 0.060 | - | - | 0.095 | 0.060 | - | - |
| Oat | - | - | 0.530 | 0.100 | - | - | 0.530 | 0.100 |
| Corn | - | - | - | - | - | - | - | - |
| VMP | - | - | - | - | 0.014 | 0.019 | 0.017 | 0.023 |
| Nutritional characterization | | | | | | | | |
| DM (g) | 1786 | 1781 | 1783 | 1777 | 1800 | 1800 | 1800 | 1800 |
| CP (g) | 124.87 | 120.00 | 132.00 | 144.71 | 124.87 | 120.00 | 132.00 | 144.71 |
| ME-ruminants (MJ) | 13.00 | 12.99 | 14.64 | 14.64 | 13.00 | 12.99 | 14.64 | 14.64 |
| NEM (MJ) | 7.30 | 7.30 | 8.42 | 8.47 | 7.30 | 7.30 | 8.42 | 8.47 |
| Ca (g) | 9.33 | 14.16 | 9.81 | 10.54 | 9.33 | 14.16 | 9.81 | 10.54 |
| P (g) | 3.20 | 3.00 | 3.53 | 3.72 | 3.20 | 3.00 | 3.53 | 3.72 |
| Na (g) | 5.73 | 6.58 | 5.42 | 6.49 | 5.73 | 6.58 | 5.42 | 6.49 |
| Mg (g) | 2.50 | 2.86 | 2.61 | 2.50 | 2.50 | 2.86 | 2.61 | 2.50 |
| Vit.A (IE) | - | - | - | - | 8595 | 11036 | 10940 | 14121 |
| Vit.D (IE) | - | - | - | - | 1029 | 1346 | 1273 | 1687 |
| Vit.E (mg) | - | - | - | - | 39.00 | 46.00 | 55.00 | 65.00 |
| Zn (mg) | - | - | - | - | 62.84 | 82.37 | 77.60 | 103.05 |
| Fe (mg) | - | - | - | - | 80.00 | 104.44 | 99.48 | 131.29 |
| Mn (mg) | - | - | - | - | 47.45 | 66.97 | 51.81 | 77.26 |
| Cu (mg) | - | - | - | - | 10.50 | 14.40 | 12.05 | 17.14 |
| Co (mg) | - | - | - | - | 0.20 | 0.25 | 0.27 | 0.33 |
| I (mg) | - | - | - | - | 0.67 | 0.77 | 0.98 | 1.11 |
| Se (mg) | - | - | - | - | 0.26 | 0.33 | 0.33 | 0.42 |

VMP = vitamin–mineral premix; DM = dry matter; CP = crude protein; NEM = Net energy milk.

### 2.3.1. General Conditions of the Study

Throughout the entire research period, both batches of rams received appropriate exercise and feeding regimes with similar conditions. Starting from MP, all rams from L2 were accommodated in a separate enclosure; equipped with a roof to limit the effects of solar radiation. Additionally, an air ventilation system was installed in the enclosure to reduce the ambient temperature by approximately 3 °C compared to the thermal level in the accommodation space of L1. Temperature was monitored throughout the preparation and mating periods using a thermometer/hygrometer, specifically the Alanlog, type 76113, which allowed temperature monitoring between −25 °C and +55 °C and humidity ranging from 0% to 100%.

Throughout the research, the fertility of rams was assessed to observe how can it be influenced by the additional administration of a VM (vitamin and mineral) complex, starting from SM.

### 2.3.2. Body Condition Score and the Behavior Evaluation

Body condition score (BCS) was assessed by palpating muscular mass and fat deposits on the dorsum (upper part of the trunk, loin, and rump). It was graded on a scale from 1 (thin) to 5 (very fat), with half-point increments, using a method developed by Jefferies [18]. BCS was evaluated by two experienced graders who reached a consensus score. Live weight (LW) was measured using an electronic scale with an accuracy of ±100 g.

The behavior of the rams was studied during the mating period using the method described by Goshme [19]. Based on how the rams reacted in the presence of ewes, they were rated on a scale from one to five, according to the following criteria:

Excellent (5): Typical behavior; when introduced into the testing pen, the ram is eager to breed with the ewe, restless, and challenging to restrain, showing a strong desire to mate immediately.

Very Good (4): Optimal sexual reflexes; the ram shows a desire to breed within a maximum of 3 min, although it can be challenging to restrain.

Good (3): The ram sniffs around upon introduction to the pen, performs breeding within the first 5 min, and is relatively easy to control.

Poor (2): The ram sniffs the ewe for a few minutes, and the time taken for breeding extends to 7–10 min. It is more docile and easier to handle.

Very poor (1): Displays delayed and incomplete reflexes, with a breeding time extending to approximately 10 min.

The study of behavior began at the end of the SM period and concluded at EM. The study of sexual reflexes was based on the behavioral manifestations exhibited by the rams during the mating period with the assigned females.

The behavior was assessed individually when bringing the males for mating. The reaction, the desire to mount, the time of approach, and the attitude over the female exhibiting ovulatory heat were observed.

### 2.4. Semen Analysis

The assessment of the impact of experimental factors on the quantity and quality of semen was conducted on samples collected by experienced individuals using the artificial vagina method in the following three distinct stages from both groups, namely: mount preparation (MP), start mounting (SM), and end of mating (EM). Before collection, the foreskin was cleaned to prevent contamination of the sperm. The technique was performed alternately once a day (Figure 2a) for four consecutive days at the beginning of each phase (MP, SM, and EM).

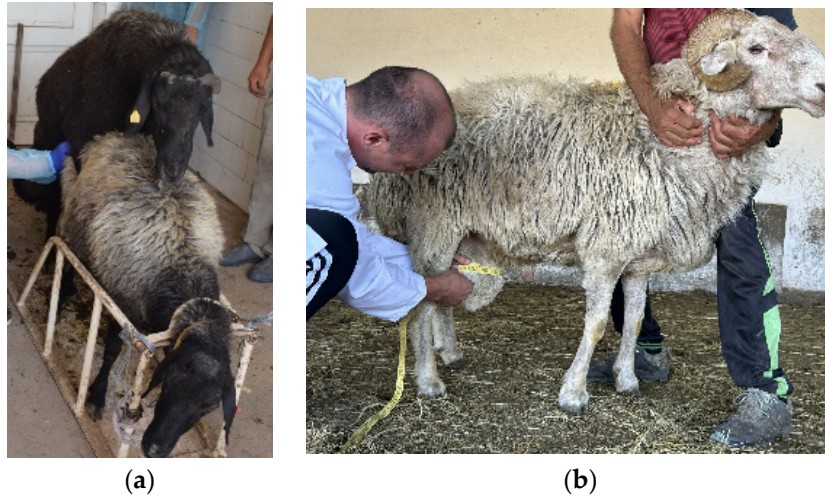

(**a**)　　　　　　　　　　　　　　　(**b**)

**Figure 2.** Semen collection (**a**) and scrotal circumference measuring (**b**).

To ensure an even distribution of the time interval between two collections, on the first day, seminal material was collected at 8 a.m. and on the following day at 8 p.m. The collected samples were transported to the laboratory within a maximum of 10 min and stored in a water bath at 37 °C.

### 2.4.1. Scrotal Circumference (SC)

The scrotal circumference was measured for each male during the collection activities by firmly pulling the testicles down towards the lower part of the scrotum and then placing a measuring tape around the widest point (Figure 2b).

### 2.4.2. Ejaculate Volume, Color, and pH

The collected volume was measured directly on the graduated flask used for collection, and the color of semen was visually assessed, giving grades from 1 to 4 following a model presented by Jha et al. [20], where the color varied from 1 to 4 (1 = watery, 2 = milky, 3 = yellowish white, 4 = creamy white).

The pH was estimated using an indicator paper (phenolphthalein paper) by matching it against a color scale.

### 2.4.3. The Evaluation of Specific Characteristics of Semen

Each ejaculate was analyzed separately, and the characteristics that determine the biological quality of the semen were determined. The analysis of specific seminal fluid traits was performed using the Computer-Aided Sperm Analysis System (CEROS II CASA, IMV Technologies, L'Aigle, France), equipped with a high-resolution Hamilton Thorne video camera with a resolution of $782 \times 582$ and progressive scanning and Trinocular Zeiss Axiolab A5 microscope (Carl Zeiss Microscopy GmbH, Jena, Germany) and the Windows 10 operating system, and report system Software HT CASAII—Designer, v. 720271 (Hamilton Thorne Inc., Beverly, MA, USA). This system is reliable and user-friendly because the CEROS II processor is separate from the optical component, allowing for precise, repetitive, and automated evaluation of the most important specific traits of seminal fluid. The video camera was connected to the microscope through an adapter, enabling video recordings at a resolution of $800 \times 600$ pixels and 60 frames per second. Sperm morphology was evaluated at a magnification of $200–400\times$ using phase contrast.

### 2.4.4. Assessment of Blood Testosterone Level

To determine the testosterone level, blood samples were collected in the first three days of the MP, SM, and EM periods. The blood samples were collected at 8:00 a.m. by making an incision in the jugular vein of each ram (a total of 135 blood samples per group). Blood collection was performed in individual tubes. After collection, the samples were transported to the laboratory, centrifuged, and the obtained serum was frozen at $-20$ °C until testosterone testing [21]. For testosterone determination, the radioimmunoassay (RIA) method was used, employing TESTO-RIA-CT (DIAsource ImmunoAssays, Ottignies-Louvain-la-Neuve, Belgium) and a commercial kit (KIP1709) with a testing range from 0.1 ng/mL to 17 ng/mL and a sensitivity of 0.05 ng/mL [22,23].

### 2.5. Statistical Data Processing

Experimental data were input in column type database, submitted to processing within the GraphPad Prism, v. 9.4.1. software (Palo Alto, CA, USA) to obtain statistical descriptors values (mean, standard deviation) and to compare the two groups' performances, using the unpaired *t*-test followed by Welch correction, assuming the standard deviations of the groups were not equal.

## 3. Results

### 3.1. Effect of Feed Used on Live Weight (LW) and Body Condition Score (BCS) of Rams

During a complete cycle (from December 2021 to December 2022), even though the maintenance condition of the rams undergoes certain changes, it is essential that during the MP period the factors that have a real influence on their reproductive condition are optimized. During the resting periods, both groups were fed the same diet, which included seasonal forage (Table 1). If the crude protein in the daily diet was provided at an average level of 122.4 g during the rest period, then in the preparation period the protein level increased by 7.84%, reaching a level of 144.71 g during the sexual activity period (Table 1).

Through the daily diet, the aim was to provide an average energy level of 13 MJ ME-ruminants during the resting period, and 14.64 MJ ME-ruminants for the preparation and intense sexual activity period. The energy level for the rest period was determined based on the animals' body weight at the maintenance level, using the animal nutrition optimization program HYBRIMIN Futter 5. For the preparation for breeding and during the mating period, the energy level was slightly higher (+12.5%).

According to results in Table 2, the body condition score significantly improved (+15.9%, $p < 0.05$) in L2 rams, compared to L1 ones and the live weight differed also in a significant manner (+4.7%, $p < 0.001$) due to stimulative feeding.

**Table 2.** Descriptive statistics for BCS (points) and live weight (kg) for rams.

| Assessment Moment | Trait | L1 | | | L2 | | |
|---|---|---|---|---|---|---|---|
| | | **Mean** | **±St. Dev** | **CV%** | **Mean** | **±St. Dev** | **CV%** |
| Preparation for mating (MP) | BCS (points) | 2.50 | 0.42 | 16.80 | 2.70 | 0.32 | 11.85 |
| | LW (kg) | 87.49 | 2.39 | 2.73 | 87.92 | 2.31 | 2.63 |
| Start mating (SM) | BCS (points) | 2.83 | 0.56 | 19.79 | 3.23 | 0.53 | 16.41 |
| | LW (kg) | 89.09 | 2.91 | 3.27 | 90.49 | 2.42 | 2.67 |
| End of mating (EM) | BCS (points) | 2.33 [a] | 0.41 | 17.60 | 2.70 [b] | 0.49 | 18.15 |
| | LW (kg) | 86.06 [a] | 2.16 | 2.51 | 90.13 [d] | 2.19 | 2.43 |

BCS = body condition score; LW = live weight. Means with different superscripts within the same row signify: [ab] = $p < 0.05$; [ad] = $p < 0.001$

### 3.2. Physical Evaluation of Reproductive Quality

Statistical data processing indicates small variations between the two groups for SC at the MP, with average values ranging between $31.45 \pm 0.56$ cm for L1 and $31.25 \pm 1.76$ for L2, and these differences are not statistically significant. The average coefficient of the variation value indicates a certain level of homogeneity, which is due to the fact that rams from both groups were carefully selected for reproduction.

However, the data obtained after measuring SC at the start of mating (SM) show a significant difference between the groups at $p < 0.05$ and a very significant difference in EM at $p < 0.001$.

The volume of ejaculate (Ev) exhibited variations with different statistical significance during the measurement period. While at the beginning of MP (mount preparation), the difference between the two groups was minimal and not statistically significant, higher values were obtained in L2 for the samples collected during SM (start mounting) and EM (end of mating) (Table 3). In this group, Ev was superior by 17.78% at the time of SM and 10.20% at EM compared to the same traits in L1.

The color of semen varied quite differently between the two groups. At the beginning of the MP period, the baseline color was milky white with shades ranging from white to creamy in both groups, and the differences between the groups were not statistically significant. However, after the experimental treatment and sampling during SM and EM,

the color of ejaculate in the rams from L2 intensified and took on shades towards creamy white, with the difference between the groups being highly significant ($p < 0.001$).

**Table 3.** Descriptive data for some physical assessments of rams and semen characteristics.

| Assessment Moment | Trait | L1 | | | L2 | | |
|---|---|---|---|---|---|---|---|
| | | Mean | ±St. Dev | CV% | Mean | ±St. Dev | CV% |
| Preparation for mating (MP) | SC (cm) | 31.45 | 1.56 | 4.96 | 31.25 | 1.76 | 5.63 |
| | Ev (mL) | 1.20 | 0.10 | 8.33 | 1.25 | 0.09 | 7.20 |
| | Color (1–5 degrees) | 2.733 | 0.594 | 6.57 | 2.867 | 0.516 | 8.14 |
| | pH | 7.033 | 0.188 | 7.31 | 7.027 | 0.166 | 8.14 |
| Start of mating (SM) | SC (cm) | 33.8 [a] | 1.83 | 5.40 | 35.42 [b] | 1.72 | 4.01 |
| | Ev (mL) | 1.30 [a] | 0.11 | 8.46 | 1.58 [d] | 0.24 | 11.19 |
| | Color (1–5 degrees) | 2.846 [a] | 0.520 | 9.22 | 3.867 [d] | 0.352 | 8.31 |
| | pH | 7.058 | 0.085 | 9.23 | 7.100 | 0.093 | 8.38 |
| End of mating (EM) | SC (cm) | 32.10 [a] | 1.80 | 5.61 | 34.71 [d] | 1.34 | 3.86 |
| | Ev (mL) | 1.32 [a] | 0.18 | 13.64 | 1.47 [b] | 0.20 | 13.61 |
| | Color (1–5 degrees) | 2.733 [a] | 0.458 | 10.24 | 3.267 [d] | 0.458 | 9.01 |
| | pH | 7.020 | 0.094 | 10.24 | 7.060 | 0.074 | 9.05 |

SC = scrotal circumference; Ev = ejaculate volume. Means with different superscripts within the same row signify: [ab] = $p < 0.05$; [ad] = $p < 0.001$.

The reaction of semen (pH), or the acidity level, is assessed by analyzing the pH value. The determinations (Table 3) on the samples collected at the respective stages indicated average values ranging from a minimum pH of $7.02 \pm 0.094$ (L1 for EM samples) to a pH of $7.100 \pm 0.093$ (L2 for SM samples), with no significant differences recorded between the groups.

### 3.3. Sperm Quality Analyses: Concentration, Live Spermatozoa, Abnormal Spermatozoa, Mobility, Testosterone

Sperm concentration determined in the samples collected in each of the three periods showed that in all situations, the average values were $>3.0 \times 10^9$ mL$^{-1}$. The highest values for the total number of sperm per unit volume (Table 4) were recorded in the samples collected at SM and EM. The fact that in the samples collected from L2, the concentration increased from $4.01 \pm 0.08 \times 10^9$ to average values of $4.41 \pm 0.12 \times 10^9$ supports the beneficial effect of the additional administered VM complex and its contribution to the increased biological value of seminal fluid. In both cases, the difference between the groups in the samples collected at SM and EM is highly significant ($p < 0.001$).

Live spermatozoa are perhaps one of the most important traits on which the male's fertility depends. Determining this trait in the samples collected at the beginning of the MP (mount preparation) period reveals non-significant differences ($p > 0.05$) between the groups. However, at the EM (end of mating) stage, the proportion of live spermatozoa is higher by 6.86% in L2, with the difference being highly significant ($p < 0.01$).

Abnormal spermatozoa, if present in a large number, can affect fertility, and unfortunately, this deficiency cannot be detected unless the quality of the seminal material is evaluated. The rate of anomalies recorded in the semen is roughly similar in both groups but occurs at different levels. In L1, during the MP, there is a reduction from $3.70 \pm 0.15$ to $3.59 \pm 0.12$ in the MP samples. In L2, it is observed that after the MP period, there is a reduction in the proportion represented by abnormal spermatozoa from $3.78 \pm 0.10$ to $3.53 \pm 0.16$. When evaluating abnormal spermatozoa in the samples collected at EM, the

average values range between the proportion limits of 3.82 ± 0.12 in L1 and 3.57 ± 0.06 in L2.

**Table 4.** Evaluation of sperm quality and serum testosterone.

| Assessment Moment | Characteristic | L1 | | | L2 | | |
|---|---|---|---|---|---|---|---|
| | | **Mean** | **±St. Dev** | **CV%** | **Mean** | **±St. Dev** | **CV%** |
| Mount preparation (MP) | Sperm concentration ($\times 10^9$/mL) | 3.98 | 0.08 | 2.01 | 4.01 | 0.08 | 2.00 |
| | Live spermatozoa (%) | 80.53 | 2.29 | 2.84 | 80.73 | 1.10 | 1.36 |
| | Abnormal spermatozoa (%) | 3.70 | 0.15 | 4.05 | 3.78 | 0.10 | 2.65 |
| | Mobility (%) | 83.87 | 5.05 | 6.02 | 85.20 | 4.07 | 4.78 |
| | Testosterone (ng/mL) | 2.29 | 0.09 | 3.39 | 2.31 | 0.27 | 11.69 |
| Start mounting (SM) | Sperm concentration ($\times 10^9$/mL) | 4.13 [a] | 0.17 | 4.12 | 4.41 [d] | 0.13 | 2.95 |
| | Live spermatozoa (%) | 81.87 [a] | 1.77 | 2.16 | 87.80 [d] | 1.93 | 2.20 |
| | Abnormal spermatozoa (%) | 3.59 | 0.12 | 3.34 | 3.53 | 0.16 | 4.53 |
| | Mobility (%) | 85.93 [a] | 4.40 | 5.12 | 88.87 [b] | 2.29 | 2.58 |
| | Testosterone (ng/mL) | 2.47 [a] | 0.11 | 4.45 | 3.18 [d] | 0.27 | 8.49 |
| End of mating (EM) | Sperm concentration ($\times 10^9$/mL) | 3.46 [a] | 0.20 | 5.78 | 3.87 [d] | 0.19 | 4.91 |
| | Live spermatozoa (%) | 79.60 | 0.99 | 1.24 | 81.67 | 2.26 | 2.77 |
| | Abnormal spermatozoa (%) | 3.82 [a] | 0.12 | 3.14 | 3.57 [c] | 0.06 | 2.68 |
| | Mobility (%) | 85.13 [a] | 2.61 | 3.07 | 88.20 [c] | 2.01 | 2.28 |
| | Testosterone (ng/mL) | 2.45 | 0.13 | 5.31 | 2.68 | 0.13 | 4.85 |

Means with different superscripts within the same row signify: [ab] = $p < 0.05$; [ac] = $p < 0.01$; [ad] = $p < 0.001$.

Spermatozoa mobility is an important factor that ensures the movement of sperm through the female reproductive tract after sexual contact. Typically, when mobility is weak, it is associated with low fertilization rates in many mammal species, including rams. According to the obtained data, it is observed that mobility had very good values in all analyzed stages. This is due to the fact that the food provided to all rams met their daily nutritional requirements, and the provided sexual rest was balanced.

At the MP stage, no significant differences are observed between the two groups. However, after the introduction of VM supplements into their diet, mobility recorded higher values in L2 (Table 4). In the samples collected at SM, which is after a 60-day period during which the rams in L2 received VM supplements, mobility differed between the groups, being 87.80 ± 1.93% in L2 and 81.87 ± 1.77 in L1, with the difference being significant ($p < 0.05$).

Testosterone levels decreased in L1 during the MP and SM intervals, while in L2, they increased from the beginning of the preparation period (MP) from 2.31 ± 0.27 ng/mL to 3.18 ± 0.27 ng/mL. At the start of the reproductive activity, the testosterone level was highly significant ($p < 0.001$).

### 3.4. Behavior Type and Fertility of Rams

The study of behavioral patterns during the SM period was based on a realistic assessment of the manifestation of typical reflexes, the desire to mount, and the average time that elapsed from introduction into the pen until the mounting took place (Table 5).

**Table 5.** The type of behavior in rams.

| Manifested Behavior | Grade | L1 | | L2 | |
|---|---|---|---|---|---|
| | | n | % | n | % |
| Excellent—Typical behavior; eager to breed upon introduction into the testing pen, restless, challenging to restrain, showing an immediate desire to mate. | 5 | 3 | 20.00 | 9 | 60.00 |
| Very good—Optimal sexual reflexes, the ram shows a desire to breed within a maximum of 3 min, although it can be challenging to restrain. | 4 | 6 | 40.00 | 4 | 26.66 |
| Good—Upon introduction to the pen, the ram sniffs around, breeding occurs within the first 5 min, and it is relatively easy to control. | 3 | 4 | 26.66 | 2 | 13.34 |
| Poor—Sniffs the ewe for a few minutes, the time taken for breeding extends, the ram is more docile, and easier to handle. | 2 | 2 | 13.34 | 0 | 0 |
| Very poor—Displays delayed and incomplete reflexes, mounting time extends to approximately 10 min. | 1 | 0 | 0 | 0 | 0 |
| Total | | 15 | 100 | 15 | 100 |
| Statistical data (Mean $\pm$ Stdev) | | 3.66 [a] $\pm$ 0.95 | | 4.46 [b] $\pm$ 0.55 | |

Means with different superscripts within the same row, between groups, signify: [ab] = $p < 0.05$; n = the number of rams; grade = the grade offered when the rams' behavior is evaluated.

Data analysis indicates the existence of an uneven distribution of the number of rams that received scores associated with good reproductive quality, higher than 3 points. The average score obtained by L2 was approximately 17.937% higher than L1, and it was statistically significant ($p < 0.05$).

The fertility of rams is an important aspect in sheep farming activities. Ram fertility is influenced by seasonal factors, and only those that undergo an adaptation period can achieve very good results. Analyzing this trait is crucial because it depends on both the number of mounted and pregnant females and the number of lambs obtained from each ram used for reproduction. The fact that each ram in L2 mated more females than those in L1 (+10.10%) and that the number of lambs obtained at birth was higher by 12.27% in L2, compared to L1, confirms that the fertility of the rams was influenced by the experimental factors (Table 6).

The higher value of fertility in the second batch (L2) is confirmed by the fact that the rams in this batch mated with approximately 93.10% of the total number of ewes assigned in the breeding program for the Botoșani Karakul sheep breed, while those in the first batch (L1) mated with only 83.69% of the total number of ewes.

**Table 6.** Ram fertility in relation to mating ewes and the lambs obtained.

| | | Pregnant Ewes | | | Lambs Obtained | | |
|---|---|---|---|---|---|---|---|
| Batch | Mean | $\pm$St. Dev | CV% | % of Mounted Ewes | Mean | $\pm$St. Dev | CV% |
| L1 | 29.29 [a] | 2.29 | 7.78 | 83.69 | 29.41 [a] | 2.24 | 6.46 |
| L2 | 32.59 [d] | 1.78 | 5.57 | 93.10 | 33.53 [d] | 1.67 | 5.15 |

Means with different superscripts within columns, between groups, signify: [ad] = $p < 0.001$.

### 3.5. The Effect of the VM Supplement on Economic Indicators

Since the research was conducted in a research station that also serves as a technology transfer facility for other farmers, an economic analysis was performed for the main

production indicators (Table 7). The accounting values reveal the fact that all costs related to maintenance, feeding, electricity, and additional VM administration were only 6.18% higher in Batch 2 (L2).

**Table 7.** Economic aspects and variable total costs.

| Economic Indicators | Measure Unit | L1 | L2 | The Difference between Groups | |
|---|---|---|---|---|---|
| | | | | ± | % |
| Fertilized ewes | n | 440 | 489 | 49 | +10.02 |
| Obtained lambs | n | 441 | 502 | 61 | +12.15 |
| Costs/ram/day | € | 3.25 | 3.36 | 0.11 | +3.27 |
| Costs/ram/year | € | 1186.25 | 1264.40 | 78.15 | +6.18 |
| Income from lambs | € | 33,075 | 37,650 | 4575 | +13.8 |
| Income per ram | € | 2205 | 2510 | 305 | +13.8 |

n = number of rams; L1 = group 1, L2 = group 2.

However, these additional costs that resulted from the experimental factors that were applied are compensated by the economic indicators, such as the increased number of ewes mated in Batch 2 (+10.02%) and a higher number of born lambs (+12.15%).

While the cost per ram per year increased by 6.18% due to specific feeding investments in the L2 group, the generated income also increased to a greater extent (13.8%) when compared to the L1 control group. This justifies the use of stimulating feeding in breeder rams.

## 4. Discussions

### 4.1. Live Weight (LW) and Body Condition Score (BCS)

The determination of LW (live weight) performed at the beginning of the MP and SM periods indicates that the differences between the batches are small and statistically insignificant ($p > 0.05$), suggesting that the VM supplements did not cause significant variations. However, it is worth mentioning that at the end of the season (EM), rams in Batch 2 had a higher LW by 4.7 kg compared to Batch 1. This can be explained by the better mobilization of body reserves in Batch 2 during the previous periods and the contribution of the additional VM complex. Therefore, after the end of breeding season (EM), the reduction of body weight was better managed by the rams in Batch 2, as the difference in LW was superior and highly significant ($p < 0.001$).

In practice, these differences are due to the fact that the diet for batch L2 was aimed at providing an additional level of certain bioactive and semen-forming substances. Energy and protein deficits are well-known causes of reproductive disorders in domestic animals [10]. The supplementation of the diet in batch L2 had a positive effect and better supported the effort during mating. Under similar conditions but with the provision of an additional daily VM diet, it was observed that in L2, live weight increased by only 2.8% from the beginning of mating, while in L1, it increased by only 1.8%. The increased energy and crude protein facilitated the replenishment of body reserves, having a positive effect on BCS (Body Condition Score), as during the preparation period for mating, the body condition improved from 2.50 points to 2.83 points in L1. In L2, the addition of bioactive substances to the diet not only led to higher LW but also improved BCS, as a BCS of ≥3 points were achieved at the beginning of the SM period.

What is interesting is that although the rams had the same mating program, the loss of live body mass due to the effort made during the mating period is different. While at the beginning of the mating period (MP), the live weight (LW) between the batches was relatively close (87.49 ± 2.39 kg in L1 and 87.92 ± 2.31 kg in L2), at the end of the mating

period, there is a difference of 4.48%. The evaluation of the body condition at the end of the mating period (EM) indicates a significant difference ($p < 0.05$) between the two groups.

In the case of rams, the lack of mineral and vitamin complexes (MV) can lead to the onset of pathologies that will affect reproductive activity, frequently affecting fertility. Similarly, feeding rations with excess salt can have negative effects on growth rates in young animals and on the reproductive function of rams and ewes, as it can lead to hormonal imbalances, such as testosterone, FSH, LH, and leptin [24].

The results obtained from the evaluation of SC (Table 3) indicate a certain evolution in the mean values obtained during the period when these determinations were made. Considering that both groups of rams received similar conditions throughout the research, we can conclude that these trends in mean values for the analyzed traits are due to the experimental treatment applied only to L2. The increased SC in the L2 rams suggests better support for the spermatogenesis process, with a positive effect on ram fertility. Additionally, at L2 during the SM, the average SC value was higher by 4.57%, which also led to a 10.20% increase in the volume of seminal fluid collected at that time.

According to these data, the annual evaluation of breeding rams should include an SC assessment because the size of the scrotum is considered an important trait in selecting breeding rams [25]. This statement is supported by the fact that in other research, a positive genetic correlation was found between scrotal circumference and semen volume, as well as sperm mobility [26].

The volume of ejaculate (Ev) was one of the traits determined immediately after collection. The statistical analysis of the data shows variations in the volume during the three collection periods. Regardless of the collection moment, the highest average values were obtained from the L2 rams (Table 3). While the differences between groups were very small and not significant during the MP phase, in the SM period, average values of $1.58 \pm 0.24$ mL for L2 and $1.30 \pm 0.11$ mL for L1 were recorded, with a highly significant difference ($p < 0.001$). At the onset of the breeding season (MP), the volume increased by over 17.72% in L2, and the difference between groups was of the same level of statistical significance ($p < 0.001$). At the end of the reproductive season (EM), based on samples collected in the last days, a significant difference ($p < 0.05$) in Ev was observed between the two groups. Essentially, the differences in semen volume were supported by the additional VM complex administered to L2.

Similar data have been reported by other research groups. For example, in research conducted on sheep in Mexico, results confirmed that supplementing the daily diet with organic zinc (Zinpro Performance MineralsTM) at a rate of 70 ppm/day above the basal diet (containing 19.6 ppm Zn) significantly influenced not only sperm concentration but also the volume, which increased from 0.69 mL to 0.97 mL [27].

The color was a distinct element, and the final data indicate an intensification of color in L2 and the appearance of different color shades between the groups. This change in the basic color and the appearance of several modifications lead to very significant differences ($p < 0.001$) and are due to the increased sperm concentration. Between the groups, the color of the sperm showed a non-significant variation in samples collected during MP ($p > 0.05$), with average scores of $2.733 \pm 0.594$ for L1 and $2.867 \pm 0.516$ for L2 (from yellowish to slightly creamy). Samples evaluated after collection from SM and EM showed that L2 had semen with a color associated with good quality (from milky white to creamy). This indicates that the administration of VM more effectively supported the spermatogenesis process. In both assessments, it was found that the differences between the groups were highly significant ($p < 0.001$).

Furthermore, the appearance of these differences also indicates a much better quality of semen color in L2 as a result of the VM supplements that significantly improved many of the seminal material traits, including color. Similar data have been reported in other research as well. For instance, in Bangladeshi rams, the color of semen varied significantly ($p < 0.05$) among rams, ranging from $1.9 \pm 1.0$ to $4.0 \pm 0.0$ (from yellowish to creamy). Some rams significantly ($p < 0.05$) exhibited semen of good quality (from milky white to

creamy), while others significantly ($p < 0.05$) presented atypical color after undergoing the same experimental design [28].

pH as an indicator of sperm acidity or alkalinity, is an important trait in the analysis of semen quality in mammals. The obtained mean values fall within relatively narrow ranges, and there are no significant differences between the two groups. The pH of the seminal fluid remained at a constant level very close to pH = 7, a level that falls within the variability limits cited in other bibliographic sources. For the pH of semen determined in various ram breeds, the range of variation is reported to be between 5.9 and 7.3 [29,30]. In the case of Dorper ram groups, the mean values for pH semen ranges from $6.7 \pm 0.3$ to $6.9 \pm 0.2$ for samples collected with an artificial vagina and also for those collected using an electro-ejaculator [31]. From these data, it can be assumed that using an electro-ejaculator during collection did not overly stimulate the accessory glands, which would normally lead to a more alkaline semen pH [29].

The mean values obtained for both groups indicate that the pH falls within the cited limits and those determined in Karakul Botoșani sheep breed [32]. The fact that semen pH is at a level that supports the content of epididymis and accessory gland secretions at an almost constant level is eloquent evidence that the spermatogenesis process is proceeding normally, with mean values at the same level in any of the three periods, and the observed differences were not statistically significant ($p > 0.05$).

### 4.2. The Quality of Sperm

For a real evaluation, at least two, and occasionally three, semen analyses are necessary, each obtained after a certain interval of several weeks. Only in these conditions can conclusive data be obtained to certify the semen quality of a male [33]. In line with this suggestion, in the present research, it was decided that all determinations be made at different times (MP, SM, and EM) placed in the vicinity of the period in which rams are prepared for mating.

Sperm concentration is influenced by the diet and the additional administration of bioactive substances in L2. The determinations made on the collected samples indicate a highly significant increase ($p < 0.001$) in the number of sperm in L2 in the samples collected during SM and EM. In samples collected during MP, the difference between the two groups is only 1%, and it has no statistical significance ($p > 0.05$). This can be explained by the fact that both batches received the same experimental treatment until the onset of MP. Therefore, in conditions where the experimental treatment was similar until the beginning of MP, it is clear that a daily supplementation of feed with VM played an extremely important role, positively influencing not only the quantity but also the concentration of the collected seminal material during SM.

This is possible because vitamins and some mineral salts are associated with increased male fertility by providing the appropriate nutrients for the reproductive system. The role of vitamins (A, D, E, etc.) is not fully understood and still represents an area where future research will bring new findings. However, since 1972, [31] it has been demonstrated that a vitamin A deficiency in mammals induces several negative effects (inhibition of spermatogenesis, reduced testicle size, and decreased testicle volume), affecting male fertility. Currently, very few farmers supplement the daily ration of rams with carrots or other feeds rich in carotene around the breeding season. As for the role of calcium (Ca), caution is needed because excessive calcium intake can disrupt reproductive function by inhibiting the absorption of various microelements (P, Mg, Zn, Cu, etc.) in the intestines. The dietary calcium-phosphorus ratio should be at least 2:1 and should not exceed 7:1. Ratios lower than 2:1 can increase the prevalence of urinary calculi [34].

Live spermatozoa are a trait influenced by various micro and macronutrients, much like sperm production. This was the reason for using the VM complex, and the results obtained have demonstrated that the experimental design was appropriate. The proportion of live spermatozoa is quite similar at the beginning of the MP, with $80.53 \pm 2.29$ in L1 and $80.73 \pm 1.10$ in L2, with no significant differences noted. However, after the MP period, the

effect of VM leads to highly significant differences ($p < 0.001$) in the total number of live spermatozoa between the batches. By the end of the breeding season (EM), the proportion of live spermatozoa in L2 is 2.53% higher ($79.60 \pm 0.99$ in L1 and $81.67 \pm 2.26$ in L2), with the difference being significant ($p < 0.01$).

The administration of the VM complex played a significant role in achieving these significant differences. Many scientific articles affirm that the beneficial effect of certain salts and vitamins on spermatogenesis is extremely important. For example, vitamin E supports sperm production in rams primarily through its antioxidant effect. The results obtained in the research conducted on Botoșani Karakul sheep breed rams are consistent with other scientific studies where similar values have been obtained. For instance, in an extensive scientific study where rams were treated twice a week with 5 mg sodium selenite and 450 mg vitamin E for 1 month, both the quality and quantity of sperm were significantly affected by the treatments. In the end, improvements were noted in ejaculate volume and mass activity. This treatment significantly influenced ($p < 0.01$) sperm concentration, which increased in the treated rams compared to the control group, and the proportion of dead and abnormal spermatozoa was lower in the treated groups [35].

Abnormal spermatozoa represent the most serious factor affecting male fertility. A high proportion of spermatozoa showing irregularities and non-conformities significantly affects the fertility of ewes. The obtained results highlight certain differences. The fact that the average values were higher in L1 suggests that the VM complex was effective and contributed to obtaining seminal material with a considerably reduced proportion of abnormal spermatozoa. Confirmation is provided by the fact that by the end of the breeding season (EM), the proportion of abnormal spermatozoa in the seminal fluid collected from L1 was approximately 6.54% higher than in L2, and this difference was statistically significant ($p < 0.01$).

It is likely that the administration of VM supplements, particularly the introduction of vitamin A, played a major role in achieving these differences and improving fertility. The effect of vitamin A and its metabolites has been highlighted in other scientific studies, demonstrating that consistent administration of this vitamin plays an essential role in improving the reproductive performance of rams [34–37]. Among the effects associated with vitamin A administration are improvements in spermatogenesis, sperm quality, libido, and stimulation of testosterone secretion. Other scientific studies specify that vitamin E plays a crucial role because it is associated with antioxidant enzymes and thus contributes to maintaining the functional competence of sperm exposed to oxidative stress. This vitamin also contributes to increased sperm viability and reduces lipid peroxidation when subjected to oxidative stress inducers [38].

Mobility is ensuring the migration of spermatozoa through the female reproductive tract and aiding them in penetrating the ovum. When mobility is reduced, it is associated with low fertilization rates in many mammalian species, including rams. Therefore, mobility is one of the most important parameters used to assess the in vitro quality and function of ram sperm [39]. Determining the mobility of spermatozoa in samples collected during the breeding season (SM) indicates higher values in L2. The fact that the difference between the average mobility values determined in the two batches was 3.41% ($p < 0.05$) suggests that the administration of VM had a positive influence on maintaining this trait at a higher level in L2 due to the inclusion of the VM complex in their diet.

Testosterone is a hormonal product secreted by the Leydig cells in the testicles. Essentially, the entire process of spermatogenesis is based on cellular events and is dependent on the level of testosterone in the testicles. In the absence of the testosterone receptor or androgens, spermatogenesis fails to progress beyond the meiotic stage [40]. To increase ram fertility, management actions should be introduced to support an increase in the blood testosterone level during periods within the intense sexual activity season of rams.

Blood sample analysis indicates similar average concentration levels per ml during the breeding season's preparation phase (MP), with no statistical significance between batches. However, an analysis of the data obtained from samples collected during the SM phase

indicates a higher level of testosterone in the rams from L2. In L1, the average value was $2.47 \pm 0.11$ ng/mL, while in L2, it was $3.18 \pm 0.27$ ng/mL, and the difference was highly significant ($p < 0.001$). The same statistical significance was obtained in the determinations made on samples collected during the EM phase ($p < 0.001$).

The results confirm that, in the case of the L2 rams, the additional feed conditions with VM and housing in covered and ventilated spaces have facilitated a better expression of the hormonal secretion process. Our results are in line with the trend of values cited in the scientific literature, which address similar aspects. Therefore, Casao et al. in 2010 [41] concluded that throughout a calendar year, there are certain monthly variations in serum testosterone, with a decrease after the winter solstice and a minimum level reached in May (testosterone) or June and July (melatonin). After this reduction, there is an increase, with a maximum level reached during the natural autumn season ($p < 0.01$). The content of testosterone in the seminal plasma of rams reached a minimum in May, subsequently increasing to a maximum level of $35.52 \pm 8.71$ ng/mL in November ($p < 0.01$). Both melatonin and testosterone values showed a seasonal variation ($p < 0.01$), with low levels outside the breeding season and high concentrations during the natural breeding season.

The lower values recorded outside the breeding season (December to June) are likely due to the lower temperatures during the winter season. There is scientific evidence confirming that prolonged heat stress leads to a decrease in testosterone concentrations in the blood of rams, with negative effects on spermatogenesis and sexual behavior. [10,12,15].

### 4.3. Male Sexual Behavior and Fertility of Rams

The study of the sexual behavior of animals, the approach, and all the changes that precede fertilization have aroused the interest of ethologists. To achieve positive results, the ram must be young, strong, healthy, and well-nourished. These individuals are the ones actively engaged in reproduction [32,42].

The results obtained confirm the hypotheses suggested by other scientific studies, which highlight the wide variability in the type of behavior exhibited by rams. Based on the scores obtained for the evaluation of the type of behavior displayed, an average score of $3.66 \pm 0.95$ was observed for L1 and $4.46 \pm 0.55$ for L2, with a significant difference ($p < 0.05$). The results show that all the rams in L2 received scores of three points (13.33%), four points (26.66%), and five points (60%) in the behavior evaluation.

If we only analyze the group of rams that scored less than 3 points, we observe that in L1, this group represented a proportion of 13.34%, while in L2, all 15 rams exceeded this minimum score. Achieving such differences suggests that those in L2 exhibited more intense behavior associated with being very good reproducers. This claim is further supported by values obtained for other breeds and under different conditions. For example, some studies have demonstrated that behavioral and physiological responses to positive stimuli or aversive handling were not directly linked at the group level, as some individuals responded more intensively to one type of stimulus while others responded differently to another type. The perception of individual behavior is based on a relatively wide combination of exhibited behavior types, influenced by the individual temperament of each ram and how each animal responds either to isolation from or the presence of a female.

The fertility depends on the relationship between parents, namely, the number of ewes assigned to a ram. The experimental plan followed the breeding program for Botoșani Karakul sheep breed, which recommends a ratio of 1 ram to 35 ewes. This ratio falls within the range specified by other bibliographic sources, which recommend proportions of one adult ram to 50 ewes during the autumn breeding season or one ram to 25–33 ewes during the off-season. Maintaining these specific ratios does not affect ram fertility.

The virility and breeding capacity of the rams were superior in L2 because each ram mated with approximately 32 out of 35 ewes, with a highly significant difference compared to L1 ($p < 0.001$). In terms of the total number of lambs obtained from the mated ewes, a highly significant difference between the groups is observed ($p < 0.001$), as the rams in L2 produced 12.27% more lambs from the inseminated ewes.

All the data confirm that the additional supplementation of VM had a positive influence on ram fertility. Some bibliographic sources state that the inclusion of certain minerals in the diet or their reproductive applications has improved the quality of seminal fluid, ram fertility, and their mating behavior. It has enhanced the antioxidant status, increased serum testosterone levels, and reduced abnormal spermatozoa [43].

## 5. Conclusions

Rams' dietary stimulation with a supplement of liposoluble vitamins (A, D, E) and of certain micro-elements (Zn, Fe, Mn, Cu, Co, I, Se) significantly improved the overall body condition and, mostly, the reproductive performance due to improvement of ejaculate volume, of semen quality, of testosterone level, of mating behavior, and of fertility traits.

According to our findings, an income increase of almost 14% was generated by the supplemental feeding of rams, based on generated costs of about 6%, economic values accepted by sheep breeders.

**Author Contributions:** Conceptualization, C.P. (Constantin Pascal) and I.N.; methodology, C.P. (Constantin Pascal), C.P. (Claudia Pânzaru), D.S. and D.M.; validation, C.P. (Constantin Pascal), D.S. and M.A.F.; formal analysis, I.N., M.A.F. and D.M.; investigation, I.N. and M.A.F.; resources, I.N. and M.A.F.; data curation, C.P. (Constantin Pascal), D.S. and D.M.; writing—original draft preparation, C.P. (Constantin Pascal) and C.P. (Claudia Pânzaru); writing—review and editing, C.P. (Constantin Pascal) and C.P. (Claudia Pânzaru); visualization, I.N., D.S. and C.P. (Constantin Pascal); supervision, C.P. (Constantin Pascal). All authors have read and agreed to the published version of the manuscript.

**Funding:** This study was realized via partial funding from the Romanian Ministry of Agriculture and Rural Development research program ADER 2019–2022, grant no. 814.

**Institutional Review Board Statement:** The animal study protocol was approved by the Institutional Committee for Animal Ethics of the Research and Development Station for Sheep and Goat Breeding, Popăuți—Botoșani, Romania, on 4 October 2021, as specified in the Statement on Bioethics no. 941 per 4 October 2021.

**Data Availability Statement:** Detailed data that generated the results presented in this study are available on request from the corresponding author.

**Acknowledgments:** The kind contribution of the research team from Animal resources and technologies department within Faculty of Food and Animal Sciences, Iasi University of Life Sciences is kindly recognized.

**Conflicts of Interest:** The authors declare no conflict of interest.

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
