# Peer review of "Diet Influence on Sperm Quality, Fertility, and Reproductive Behavior in Karakul of Botoșani Rams"

_agriculture, doi:10.3390/agriculture13112168_

Round 1

Reviewer 1 Report

Comments and Suggestions for Authors

revision is needed, especially the material method part

Round 1

Reviewer 1 Report

Recommendation 1

Please synchronize the tittle and introduction part, especially in the study purpose. Many external factors influence the physical regulation of ram farming, but the tittle only mentions the diet effect. For example, please reframe the title or the introduction part to be more detail about the diet only, not all the related factors. For that study, aims need to be adjusted accordingly. Material and method.

Answer

To synchronize better the title with the introduction part, we proceeded in adding some changes in this paragraph. We removed an idea where the influence of temperature was presented and we replaced the information with other that are synchronized with the title.

We removed also some references and we added actual and more specific ones.

Thank you indeed!

Recommendation 2

Line 91 – 98 Ethical approval needs to be shown the letter number or specific committee who give the clearance. Or if any show the link of specific regulation regarding the handling of animal treatment, sample handling, and any related issues which ensure not put animal in discomfort.

Answer

Thank you!

Recommendation 3

Line 100- Add the gps data of the location. Switch the position of Figure b to a, and vice versa. Since figure b give bigger overview of the location, then detailed with figure a as sampling location.

Answer

We have included the GPS coordinates of the research area at line 117.

Thank you!

Recommendation 4

Line 102 as well as line 106 Add the source of daily temperature average data claim. Could be from the website or local climate data station database. Any claim must be accompanied by the data and its source.

Answer

Following this recommendation we have added the source of the annual climatic values.

Thank you!

Recommendation 5

Line 111-112 Add the gps data location for the station.

Answer

The GPS coordinates were added at line 117.

Thank you!

Recommendation 6

Line 113 The average of body weight each group (mean ± sd) better to be added.

Answer

We have added the average values of body weight at line 132.

Thank you!

Recommendation 7

Line 135 Tabel 1 presentation was confusing. The difference between L1 and L2 is only the VMP addition. It would be simple if the diet composition was tabulated in 1 table and the WVM difference in the other table. But please clearly mentioning that L1 and L2 have similar diet, the only difference is the addition of WVM as the composition shown in the table. This related to the comment of line 113, whether the diet composition given is based on the body weight or other considerations.

Thank you!

Answer

We mentioned that the difference between the batches was administrating VM starting the preparation for breeding.

Thank you!

Recommendation 8

Line 241

The energy level was based on which consideration? Nutrient requirements or other information?

Please add the information.

Statistical data indicates small variation… (provide the value of variation here, why its categorized

as small).

Answer

Am adăugat în text informațiile solicitate la această recomandare.

Thank you!

Recommendation 9

Line 259

Table 3. only white p-value, omit the….and level of significance

Answer

We have removed the column with p value and we have changed the way of pointing out the statistical significance in all the tables.

Thank you!

Recommendation 9

Line 310. Sperm mobility or motility? Please select one and consider changing in all part of the manuscript.

Answer

Thank you!

Recommendation 10

Line 322. A decline of testosterone level after SM is observed (see table 4). MP to SM testosterone level increase but decline from SM to EM. Please revise the sentence. The finding is significant p

Answer

We have changed this aspect and also we have introduced different superscripts as follows: ab for p < 0.05; ac for p < 0.01; ad for p < 0.001.

Thank you indeed!

Recommendation 11

Line 331 Table 5. is hard to interpret, due to no information of behaviour assessment in the material method. Add the assessment technique with clear grade and calculation explanations. In addition, also add the statistical analysis of the behaviour data. See line 167, for behaviour assessment, but the grade is not the same. Please adjust accordingly.

Answer

We made the requested modification and added the assessment method for the behavior of the rams. I synchronized it with the "Materials and Methods" section.

Thank you!

Recommendation 12

Line 342 Each ram in L2 mated with over 10.10% more female… this sentence did not agree with line 140 (each ram receives 35 ewes per season. As my opinion, the season is same, the number of females per male is same, but the sentence why L2 mated more female need to be adjusted. This also applied to the number of lambs obtained. Its need an explanation if the L1 or L2 group is calculated based on the ram family or per group. There are 15 rams in L1 batch means 15 group of ram family. Please explain accordingly since the beginning of the method explanation in the mating process.

Answer

We have changed several facts to clarify that each ram bred with 25 females in that season. All males in L2 mated with 10.10% more females, resulting in a higher number of lambs by 12.27%.

Thank you!

Recommendation 13

Table 7. To complete the comparison of economic value, the estimation of the output of money in obtained lamb need to be capitalized. Shows in the table. It would be better if the benefit/cost ration between batches also calculated, though the price was using estimation. But this information would be very beneficial for the farmer.

Answer

The costs for L2 were only 6.18% higher, primarily due to adding the VM in the diet. The fact that the rams in L2 mated with 489 females (compared to 440 in L1), resulting in 502 lambs (as opposed to only 441 lambs from the females mated with L2 rams), fully justified these costs, and the farmers were satisfied with these results.

Thank you indeed!

Reviewer 2 Report

Comments and Suggestions for Authors

This manuscript studied the diet influence on sperm quality, fertility, and reproductive behavior in Karakul of Botoșani rams. It is very interesting but it lacks some important details throughout the text, as informed bellow:

Abstract - It must be improved. Authors should add some more descriptions related to the animals used on the experiment and the experimental design. Also, they should inform the analyzing methods and the main results. Please provide numeric results for the most important variables found for both groups. These changes will increase the readers understanding of your study.

Introduction - This sections works fine as a literature review. Even if it give us a general view of the subject, it lack on details related to the technologies that will be tested at the present research. Authors should refine their text to introduce the subjects of their future experimental design. If you will test a diet, please justify why to do it. Also, justify the coice for the components of the diet and how they would positively affect reproduction. Please, report if this is an inedited attempt of using such components on diet (I believe no) or if they were previously tested for other animals groups. Finally, define your hypothesis and your specific objectives.

Material and Methods - Since ovine reproduction could be affected by photoperiod, please inform the geographical coordinated for the studying area, also, inform the season (dry or wet) in which experiments were conducted and the photoperiod to which animals were subjected.

-  In line 114, as well as in the abstract, what are meaning with "a complete reproductive cycle"? Does it means the spermatogenic cycle? Or are you referring to reproductive season? Please define it, as well as indicate the number of the days or months you are referring. 

- Please provide references for all the methods used as semen collection and analysis, behavior evaluation, etc.

- Please provide details related to the chamber used for the CASA analysis. Also, report or reference CASA settings.

Results - Your text on results description should be more objective and report only the main findings. Please avoid to repeat methodologies and exclude inadequate discussions. 

- Results are well reported, in genera; however, authors should avoid internal bars on the tables.

Discussion - Even if data is technically and generally discussed, this section is extremely lengthy and boring (6.5 pages). Moreover, there is no need to separate it into different subtopics. The text should naturally flow. Please, reorganize your ideas, exclude excessive literature review, and rewrite your discussion section.

- Discussion miss some biochemical explanations about how the specific components of the diet will act on the improvement of specific semen characteristics, fertility or reproductive behavior of the rams. 

Conclusions - It is vague and looks like another literature review. Authors should be more objective to report their conclusions highlighting the idea that using a supplemented diet will improve semen quality, fertility and reproductive behavior of rams. If you note, your tittle is the most adequate conclusion for your work.

Round 1

Reviewer 2

Recommendation 1

Abstract - It must be improved. Authors should add some more descriptions related to the animals used on the experiment and the experimental design. Also, they should inform the analyzing methods and the main results. Please provide numeric results for the most important variables found for both groups. These changes will increase the readers understanding of your study.

Answer

We have rewritten the abstract and included representative research findings.

Thank you indeed

Recommendation 2

Introduction - This sections works fine as a literature review. Even if it give us a general view of the subject, it lack on details related to the technologies that will be tested at the present research. Authors should refine their text to introduce the subjects of their future experimental design. If you will test a diet, please justify why to do it. Also, justify the coice for the components of the diet and how they would positively affect reproduction. Please, report if this is an inedited attempt of using such components on diet (I believe no) or if they were previously tested for other animals groups. Finally, define your hypothesis and your specific objectives.

Answer

We have made revisions to the Introduction part, removing some non-essential information and introducing relevant content that supports the research title. I have also clearly emphasized the purpose of this research.

Thank you!

Recommendation 3

Material and Methods - Since ovine reproduction could be affected by photoperiod, please inform the geographical coordinated for the studying area, also, inform the season (dry or wet) in which experiments were conducted and the photoperiod to which animals were subjected.

Answer

have specified the GPS coordinates of the research area in the Materials and Methods section. I have also indicated the season during which the experimental plan was implemented.

Thank you indeed!

Recommendation 4

-  In line 114, as well as in the abstract, what are meaning with "a complete reproductive cycle"? Does it means the spermatogenic cycle? Or are you referring to reproductive season? Please define it, as well as indicate the number of the days or months you are referring.

Answer

We mentioned at line 133 what is the length of the complete cycle period (one year long).

Thank you indeed!

Recommendation 5

- Please provide references for all the methods used as semen collection and analysis, behavior evaluation, etc.

Answer

We have added throughout the text new information regarding the methods used.

Thank you indeed!

Recommendation 5

- Please provide details related to the chamber used for the CASA analysis. Also, report or reference CASA settings.

Answer

We have added at the line 233 more details regarding this request.

Thank you indeed!

Recommendation 6

Results - Your text on results description should be more objective and report only the main findings. Please avoid to repeat methodologies and exclude inadequate discussions. 

Answer

We have removed certain information from this chapter in order to synthesize the data.

Thank you indeed!

Recommendation 7

- Results are well reported, in genera; however, authors should avoid internal bars on the tables.

Answer

We have removed the internal bars from all the tables.

Thank you!

Recommendation 8

Discussion - Even if data is technically and generally discussed, this section is extremely lengthy and boring (6.5 pages). Moreover, there is no need to separate it into different subtopics. The text should naturally flow. Please, reorganize your ideas, exclude excessive literature review, and rewrite your discussion section.

Answer

I have reorganized the presentation of the Discussions chapter by removing several paragraphs containing related values.

Thank you indeed!

Recommendation 9

- Discussion miss some biochemical explanations about how the specific components of the diet will act on the improvement of specific semen characteristics, fertility or reproductive behavior of the rams.

Answer

We did not include biochemical data because this aspect was not the focus of this research. In future studies, we may expand the investigations to include this aspect and provide relevant scientific information.

Thank you!

Recommendation 10

Conclusions - It is vague and looks like another literature review. Authors should be more objective to report their conclusions highlighting the idea that using a supplemented diet will improve semen quality, fertility and reproductive behavior of rams. If you note, your tittle is the most adequate conclusion for your work.

Answer

The conclusions were reconsidered and changed completely to highlight them in better way.

Thank you indeed!

Round 2

Reviewer 2 Report

Comments and Suggestions for Authors

Authors conducted an extensive review on the manuscript which improved it a lot; however, some minor points should be addressed.

Abstract is too short and low informative. According to the guiding for authors, it should contain 200 words. Thus, authors are advised to improve their descriptions, mainly related methodologies. Also, they are advised to provide numeric results for the variables that showed significant differences between tested groups.

Discussion - It remains extremely long and boring. Please, try to be more objective and focus on the explanations of yourself results. There is no need of large literature review sentences.

Conclusions - If the objectives are focused on enhance the reproductive performance of rams by implementing a diet that includes a VM (vitamins and minerals) supplement, your conclusion should directly respond this objective. Thus, you should highlight if a diet including VM supplement enhanced or not the reproductive performance of rams, evidencing the main parameters enhanced. 

Author Response

Honorable reviewer,

Thank you for your valuable suggestions, we complied with them and adjusted our manuscript, as following:

Suggestion 1> Abstract is too short and low informative. According to the guiding for authors, it should contain 200 words. Thus, authors are advised to improve their descriptions, mainly related methodologies. Also, they are advised to provide numeric results for the variables that showed significant differences between tested groups.

We have improved the abstracts and the significance threshold levels for the influenced parameters. Thank you!

Suggestion 2: Discussion - It remains extremely long and boring. Please, try to be more objective and focus on the explanations of yourself results. There is no need of large literature review sentences.

We have shortened the discussion section and, consequently elliminated some references titles. Therefore, the list of References was synchronised with the new citation order. Thank you!

Suggestion 3: Conclusions - If the objectives are focused on enhance the reproductive performance of rams by implementing a diet that includes a VM (vitamins and minerals) supplement, your conclusion should directly respond this objective. Thus, you should highlight if a diet including VM supplement enhanced or not the reproductive performance of rams, evidencing the main parameters enhanced. 

We have modified the conclusions section and provided the improvement levels.

Thank you indeed!